# High Frequency of *PIK3CA* Mutations in Low-Grade Serous Ovarian Carcinomas of Japanese Patients

**DOI:** 10.3390/diagnostics10010013

**Published:** 2019-12-27

**Authors:** Tomoka Ishibashi, Kentaro Nakayama, Sultana Razia, Masako Ishikawa, Kohei Nakamura, Hitomi Yamashita, Puja Dey, Koji Iida, Hiroko Kurioka, Satoru Nakayama, Yoshiro Otsuki, Noriyoshi Ishikawa, Satoru Kyo

**Affiliations:** 1Department of Obstetrics and Gynecology, Shimane University School of Medicine, Izumo 6938501, Japan; tomoka@med.shimane-u.ac.jp (T.I.); raeedahmed@yahoo.com (S.R.); m-ishi@med.shimane-u.ac.jp (M.I.); memedasudasu1103@gmail.com (H.Y.); iida@med.shimane-u.ac.jp (K.I.); satoruky@med.shimane-u.ac.jp (S.K.); 2Department of Obstetrics and Gynecology, Shimane Prefectural Central Hospital, Izumo 6938555, Japan; kuri35@spch.izumo.shimane.jp; 3Department of Obstetrics and Gynecology, Seirei Hamamatsu Hospital, Hamamatsu 4308558, Japan; satoru@sis.seirei.or.jp; 4Department of Organ Pathology, Seirei Hamamatsu Hospital, Hamamatsu 4308558, Japan; otsuki@sis.seirei.or.jp; 5Department of Organ Pathology, Shimane University School of Medicine, Izumo 6938501, Japan; kanatomo@med.shimane-u.ac.jp

**Keywords:** *PIK3CA*, *BRAF*, *KRAS*, *ERBB2*, low-grade serous ovarian tumor

## Abstract

The frequency of *KRAS/BRAF* mutations associated with low-grade serous ovarian carcinoma (LGSC)/serous borderline tumors (SBTs) in Japan is unknown. We aimed to identify genetic variations in *KRAS*, *BRAF*, *PIK3CA*, and *ERBB2* in LGSC/SBT/serous cystadenomas (SCAs) in a Japanese population. We performed a mutation analysis (by Sanger sequencing) of 33 cases of LGSC/SBT/SCA and 4 cases of LGSC with synchronous SBTs using microdissected paraffin-embedded sections. Immunohistochemistry of p53 and ARID1A was also performed. The frequency of oncogenic mutations in *PIK3CA* was 60.0% (6/10) in LGSCs, 63.6% (7/11) in SBTs, and 8.3% (1/12) in SCAs. All cases harbored wild-type *KRAS*. The frequency of *BRAF* mutations was 20.0% (2/10) in LGSCs, whereas all SBTs and SCAs harbored the wild-type allele. The frequency of *ERBB2* mutations was 30.0% (3/10) in LGSCs, 0.0% (0/11) in SBTs, and 16.7% (2/12) in SCAs. ARID1A staining was positive in all cases. p53 staining was positive in 0% (0/10) LGSCs, 9.1% (1/11) SBTs, and 0.0% (0/12) SCAs. One LGSC case had two *PIK3CA* mutations (G1633A and G3149A) in both LGSC and SBT lesions, but a *BRAF* mutation was detected only in an LGSC lesion. These results suggest that, compared with the values in Western populations (16–54%), the *KRAS* mutation frequency in LGSCs/SBTs is lower and that of *PIK3CA* mutations in LGSCs/SBTs is much higher in Japanese populations. Therefore, the main carcinogenesis signaling pathways may be different between Japanese and Western LGSCs. Molecular therapies targeting the PIK3CA/AKT pathway may be effective in LGSCs in Japan.

## 1. Introduction

Ovarian cancer is the leading cause of death owing to gynecologic malignancies in the world [1]. Recently, ovarian cancer was subdivided into two categories, Type I and Type II [2]. Type II tumors mainly include high-grade serous carcinomas (HGSCs) with *TP53* mutations and show an aggressive clinical course. In contrast, Type I tumors include low-grade serous carcinomas (LGSCs), mucinous carcinomas, and clear cell carcinomas. LGSCs are more common in younger patients and associated with chemoresistance than HGSCs. Previous reports from Western countries have indicated that LGSCs have a higher frequency of *KRAS* (16–54%) or *BRAF* (2–33%) mutations [3,4,5]. Therefore, KRAS/BRAF/ERK signaling pathways are thought to be essential in the carcinogenesis of LGSC in Europe.

However, molecular profiles of LGSC in Japanese patients have not been determined. Recently, we identified a case of LGSC with synchronous pathological precursor tissues but without either *KRAS* or *BRAF* mutations in any lesions [6]. Therefore, we speculated that LGSCs in Japanese patients might have a low frequency of *KRAS* and *BRAF* mutations, but could be associated with other oncogenic mutations. In the current study, we evaluated the prevalence of *KRAS*, *BRAF*, *PIK3CA*, and *ERBB2* mutations in Japanese LGSCs, not only clarifying the genetic drivers of these mutations but also the difference in mechanisms of carcinogenesis between Japanese and European LGSCs. Furthermore, immunohistochemistry of p53 and ARID1A was performed as a surrogate for identifying inactivating mutations in these genes.

## 2. Materials and Methods

### 2.1. Tumor Aamples

Formalin-fixed paraffin-embedded tissue samples from 10 LGSC, 17 SBT, and 12 SCA patients were analyzed in this study. The samples were retrieved from the Department of Obstetrics and Gynecology, Shimane University Hospital (Izumo, Japan), Seirei Hamamatsu General Hospital, and Shimane Prefectural Central Hospital from 2007 to 2017. Pathological diagnoses were determined by histopathologic examination of hematoxylin and eosin-stained sections. The tumors were categorized according to the World Health Organization subtype criteria, and staged according to the International Federation of Gynecology and Obstetrics classification system. All patients were treated with primary debulking surgery (i.e., total abdominal hysterectomy, bilateral salpingo-oophorectomy, and omentectomy) with or without pelvic and para-aortic lymph node dissection and adjuvant taxane and platinum combination chemotherapy. The surgical specimens from each case were reviewed by a gynecological pathologist (N.I.). This human subjects research was approved by the Ethics Committee of the Shimane University Hospital (approval no. 2004-0381), and written informed consent was obtained from all patients. The study was conducted in accordance with the tenets of the Declaration of Helsinki and Title 45 (United States Code of Federal Regulations), Part 46 (Protection of Human Subjects), effective 13 December 2001.

### 2.2. Microdissection and DNA Extraction

Ten LGSC, 11 SBT, and 12 SCA cases had sufficient tumor tissue for DNA extraction and sequence analysis. Tissue sections reviewed and marked with lines by a skilled gynecological pathologist were placed on membrane slides and counterstained with hematoxylin. Selected tumor tissues dissected in 10-mm sections under a microscope using a 24-gauge needle to obtain a high percentage of tumor cells. After 48 h of digestion with proteinase K, DNA was extracted from the microdissected samples using a QIAmp DNA Micro Kit (Qiagen, Valencia, CA, USA) according to the manufacturer’s instructions.

### 2.3. Direct Sequence Analysis

Sanger sequencing was performed on polymerase chain reaction (PCR)-amplified *KRAS*, *BRAF*, *PIK3CA*, and *ERBB2* using genomic DNA obtained from microdissected formalin-fixed paraffin-embedded tissue. We focused on analyzing exons that were reported to harbor the majority of mutations in each of the genes. The primer sequences and PCR protocol used in this study were described previously [7]. Appendix A shows sequencing primers for all exons that were sequenced in the current study. We confirmed the pathogenicity associated with each mutation using the Catalogue of Somatic Mutations in Cancer (COSMIC) [8].

### 2.4. Immunostaining of p53 and ARID1A

Loss of ARID1A expression in tumor cell nuclei was used as a surrogate for the presence of ARID1A loss-of-function mutations [9]. Similarly, p53 immunoreactivity was used as a surrogate for the presence of p53 loss-of-function mutations. The antibodies used in this study were a mouse monoclonal antibody against ARID1A (BAF250a) (Santa Cruz Biotechnology Santa Cruz, CA, USA) and mouse monoclonal antibody against p53 (clone DO-7, DAKO, Carpinteria, CA, USA). Immunohistochemistry for ARID1A and p53 was performed on tissue specimens at a dilution of 1:50 or 1:100, followed by detection using an EnVision+ System with the peroxidase method (DAKO, Carpinteria, CA, USA). The detail protocols for immunostaining and evaluation of ARID1A and P53 have been described in previous reports [9,10].

## 3. Results

All 33 ovarian serous tumors were assessed for mutations in *KRAS*, *BRAF*, *PIK3CA*, and *ERBB2*. Interestingly, all LGSC, SBT, and SCA cases showed wild-type *KRAS* variant (Table 1, Table 2 and Table 3). The prevalence of oncogenic mutation of *PIK3CA* was 60.0% (6/10) in LGSCs, 63.6% (7/11) in SBTs, and 8.3% (1/12) in SCAs. Representative histological images and nucleotide sequences in *PIK3CA* are shown in Figure 1. The prevalence of *BRAF* mutations was 20.0% (2/10) in LGSCs, whereas *BRAF* in both SBTs and SCAs were all wild-type. The prevalence of *ERBB2* mutations was 30.0% (3/10) in LGSCs, 0.0% (0/10) in SBTs, and 16.7% (2/12) in SCAs. Details of *PIK3CA*, *BRAF*, and *ERBB2* mutation types are shown in Table 1, Table 2 and Table 3. ARID1A staining was observed in all cases. Staining of p53 was found in 0.0% (0/10) of LGSCs, 9.1% (1/11) of SBTs, and 0.0% (0/12) of SCAs. Representative images of p53 and ARID1A staining are shown in Figure 2. We also analyzed the mutation status of KRAS, PIK3CA, BRAF, and ERBB2 in LGSCs with synchronous SBTs. Representative histological images and nucleotide sequences in *PIK3CA* and *BRAF* in a case with both LGSC and SBT are shown in Figure 3 and Table 4. One LGSC case had *PIK3CA, BRAF*, and *ERBB2* mutations in the LGSC lesion but not in the SBT lesion. Another LGSC case had *PIK3CA* mutations (G1633A) in both LGSC and SBT lesions and a *BRAF* mutation only in the LGSC lesion. One other case had an *ERBB2* mutation in both LGSC and SBT lesions, and another had no mutations in either lesion (Table 4).

## 4. Discussion

In the current study, wild-type *KRAS* was found in all Japanese LGSC, SBT, and SCA cases. In contrast, *BRAF* mutations were detected in 20% (2/10) of LGSCs. These findings are consistent with a recent report of a low frequency of *KRAS* and *BRAF* mutations in Chinese patients [11], suggesting that genes driving LGSC may be different in Asian and Western populations. Furthermore, in the current study, 3 out of 10 (30%) LGSC cases showed *ERBB2* mutations. Previously, we identified *ERBB2* mutations (9.5%) in Western patients [7]. The prevalence of *ERBB2* mutations in the current study was much higher than that indicated in previous reports [7,12]. Interestingly, 16.7% of *ERBB2* mutations were detected in SCAs, suggesting that *ERBB2* mutation may be an early event in LGSC carcinogenesis. Taken together, the results of the current and previous reports also suggest that carcinogenesis of Japanese LGSCs may be different from that of Western LGSCs. Furthermore, the prevalence of oncogenic mutations in *PIK3CA* in both LGSCs and SBTs was much higher in Japanese patients than in Western patients [4,7,12,13]. Appendix A shows the prevalence of *KRAS*, *BRAF*, *PIK3CA*, and *ERBB2* mutations in European LGSCs. This high prevalence of oncogenic *PIK3CA* mutations in both SBTs and LGSCs suggests that these mutation events occur early in LGSC carcinogenesis. How does this discrepancy in PIK3CA mutation occur in patients with LGSC among different studies? One possibility is that the molecular mechanism of LGSC carcinogenesis differs between Japanese and other ethnicities. Another possibility is that the sample size in the current study was small and not representative of the Japanese population as a whole. The incidence of LGSC is quite low in Japan; therefore, a large multi-institutional cohort study is needed to confirm the current findings.

To determine the significance of *PIK3CA*, *BRAF*, and *ERBB2* mutations in carcinogenesis of LGSCs, we analyzed these mutations in LGSCs with synchronous SBTs. One LGSC case had *PIK3CA*, *BRAF*, and *ERBB2* mutations only in LGSC lesions but not in SBT lesions, whereas another LGSC case had *PIK3CA* mutations in both LGSC and SBT lesions. In contrast, *BRAF* mutations were detected only in LGSC lesions, suggesting that *BRAF* mutation is an important event in the SBT to LGSC transition. These findings suggest that both *PIK3CA* and *ERBB2* mutations are important carcinogenesis events, leading from SCA to SBT and SBT to LGSC. In contrast, *BRAF* mutation may be a late event in LGSC carcinogenesis.

According to results of the current immunohistochemical study, loss of function mutations in *ARID1A* and *TP53* are not critical events in LGSC carcinogenesis in Japanese patients. However, tumor cells from one SBT case stained positive for p53 (an indication of mutant *TP53*), suggesting that this SBT is a precursor of HGSC, and that there may be a pathway from Type I to Type II ovarian cancer [14].

Based on the current findings, we hypothesize that the main oncogenic signaling pathway in Japanese LGSCs is PIK3CA/AKT, whereas that in Western LGSCs is KRAS/BRAF/ERK (Figure 4). Recently, gene panel sequencing was introduced in clinical settings in Japan [15]. Current findings may contribute to genotyped matched therapies against LGSC in Japanese patients.

Oncology research teams from Johns Hopkins University and MD Anderson Cancer Center recently reported two lines of evidence supporting an LGSC carcinogenesis pathway. Peritoneal implants of SBTs and atypical proliferative serous tumors, which are considered precancerous lesions that will lead to LGSCs, had the same *KRAS/BRAF* mutation patterns as those of primary lesions, suggesting that these are metastatic lesions arising from primary lesions [16]. In addition, *KRAS* mutations, but not *BRAF* mutations, are involved in the progression of SBT to LGSC, and *KRAS*, a *G12V* mutation, indicates a poor prognosis [17]. Recently, Emmanuel et al. reported *NRAS* mutations in 9% of LGSCs with adjacent SBTs, suggesting *NRAS* is an oncogenic driver of LGSCs [18]. Furthermore, Gershenson et al. reported that patients with *KRAS* or *BRAF* mutations had significantly better overall survival times that those with wild-type *KRAS* or *BRAF* [19]. The Gynecologic Oncology Group recently performed a phase II study in the USA of selumentinib (AZD6244, ARRY142866), a MEK1/2 inhibitor in patients with recurrent LGSC [20]. According to their report, stable disease was observed in 65% of patients, with an overall disease control rate of 81%. Furthermore, there was no correlation between *KRAS/BRAF* mutations and response rates, so the predictive value of identifying *KRAS/BRAF* mutations against MEK inhibitor as biomarkers is still unclear. However, very recently, successful genotype-matched therapies against *BRAF*, *KRAS* and *NRAS* in patients with LGSC have been reported [21,22,23]. Unfortunately, these studies were based only on a Western, not Japanese or even Asian, population. Therefore, a large multi-institutional cohort study with genetic and epigenetic analyses is needed to elucidate the carcinogenic mechanism underlying LGSC in Japanese patients.

The current study has several limitations. First, the number of samples in this study was small. A follow-up study with an increased number of subjects is ongoing. This will enable us to determine, statistically, the relationship between the mutations identified in the present study and patient outcomes. Second, we identified genetic mutations via Sanger sequencing; therefore, the kinds of gene mutations assessed were limited. Further analyses using next-generation sequencing will also be needed to determine the molecular mechanism that underlies progression to LGSC in Japanese patients.

In summary, the current findings suggest that the mutation frequency of *KRAS* in LGSCs/SBTs in Japan is lower than that in Western countries. In addition, the mutation frequency of *PIK3CA* in LGSCs/SBTs appears to be very high in a Japanese population compared to Western populations. *PIK3CA* mutation may be a main driver and *BRAF* or *ERBB2* mutation may be a sub-driver event in Japanese LGSCs. Therefore, molecular therapies targeting the PIK3CA/AKT pathway may be effective in LGSCs in Japan.

## Figures and Tables

**Figure 1 diagnostics-10-00013-f001:**
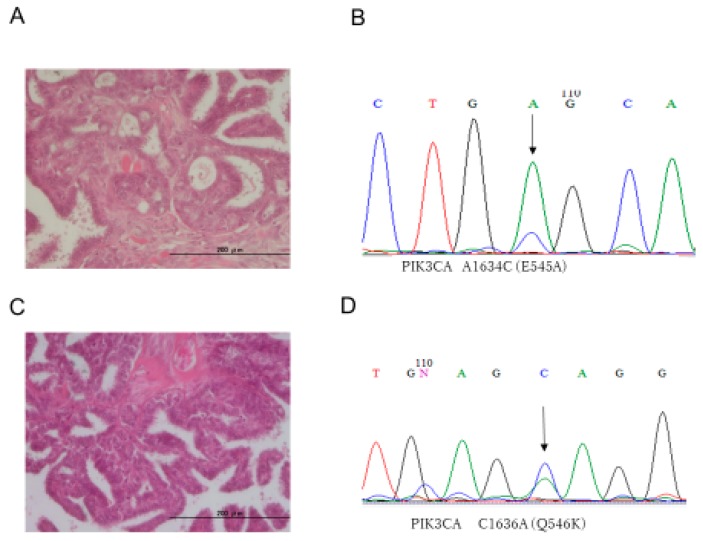
Histopathological images and nucleotide sequences of *PIK3CA* in representative LGSC and SBT cases. (**A**) Hematoxylin and eosin staining of LGSC sections. (**B**) Nucleotide sequence chromatogram showing a mutation, E545A (1634 A > C), in *PIK3CA* of an LGSC. (**C**) Hematoxylin and eosin staining of SBT sections. (**D**) Nucleotide sequence chromatogram showing a mutation, Q546K (1636 C > A), in *PIK3CA* of an SBT. Scale bar = 200 µm. C; cytosine, T; thymine, G; guanine, A; adenine.

**Figure 2 diagnostics-10-00013-f002:**
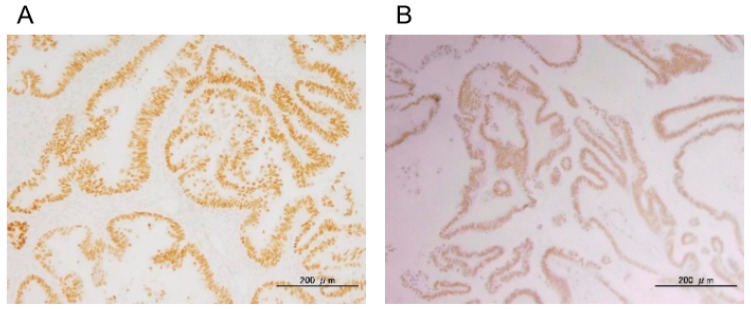
Representative positive staining of p53 (**A**) and ARID1A (**B**). Nuclear staining of p53 and ARID1A in an LGSC case.

**Figure 3 diagnostics-10-00013-f003:**
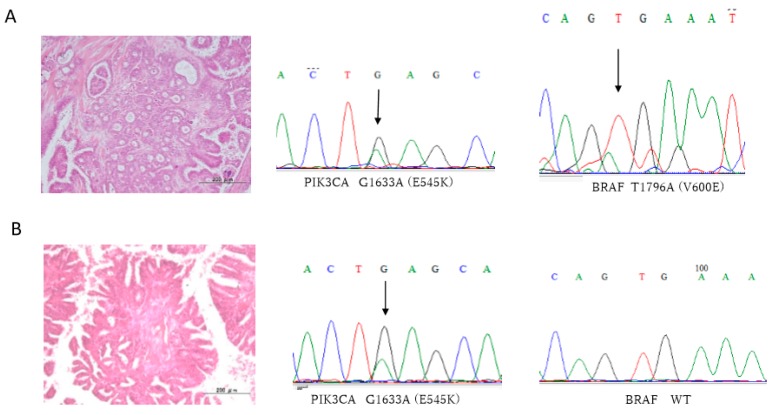
Synchronous LGSCs and SBTs with matched *PIK3CA* and *BRAF* sequences in an LGSC case. (**A**) LGSC showing mutations E545K (1633 G > A), in *PIK3CA*, and V600E (1796T > A), in *BRAF*. (**B**) SBT showing a mutation, E545K (1633 G > A), in *PIK3CA*, and wild-type *BRAF*. Scale bar = 200 µm. C; cytosine, T; thymine, G; guanine, A; adenine.

**Figure 4 diagnostics-10-00013-f004:**
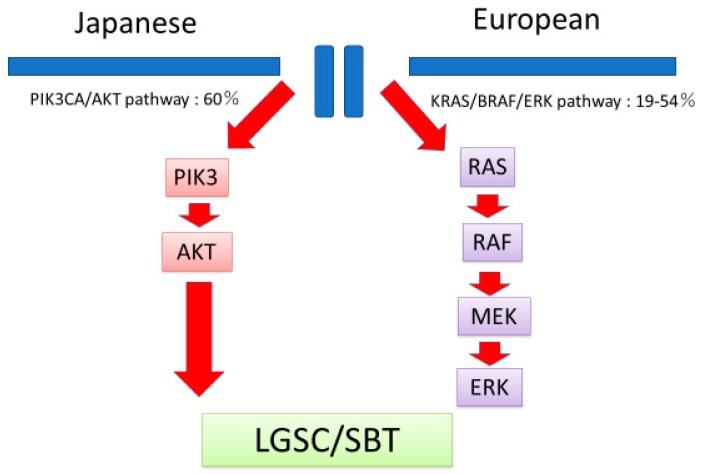
Hypothesized differences in carcinogenesis between Japanese and European patients with LGSC. LGSC in Japanese patients may depend on alterations in the PIK3CA/AKT pathway, whereas in Europeans it may depend on alterations in the KRAS/BRAF/AKT pathway.

**Table 1 diagnostics-10-00013-t001:** Mutation and imunohistochemical analysis of low grade serous ovarian carcinoma (LGSC). WT; Wild Type.

No.	Age	FIGO Stage	*KRAS*	*BRAF*	*PI3KCA E9*	*PI3KCA E20*	*ERBB2*	P53	ARID1A
1	37	II c	WT	WT	WT	WT	A2384G (Q795R)	Normal	Normal
2	61	IV b	WT	WT	G1633A (E545K)	WT	WT	Normal	Normal
3	83	I a	WT	WT	A1634C (E545A)	WT	WT	Normal	Normal
4	61	I c	WT	WT	WT	WT	A2384G (Q795R)	Normal	Normal
5	37	III c	WT	WT	G1633C (E545Q)	WT	WT	Normal	Normal
6	27	I c	WT	T1796A (V600E)	A1634C (E545A)	WT	A2384G (Q795R)	Normal	Normal
7	61	III c	WT	WT	WT	WT	WT	Normal	Normal
8	48	I c	WT	WT	WT	WT	WT	Normal	Normal
9	26	III c	WT	WT	G1633A (E545K)	WT	WT	Normal	Normal
10	40	I c	WT	T1796A (V600E)	A1634C (E545A)	WT	WT	Normal	Normal

**Table 2 diagnostics-10-00013-t002:** Mutation and immunohistochemical analysis of serous borderline tumor (SBT). WT; Wild Type.

No.	Age	FIGO Stage	*KRAS*	*BRAF*	*PI3KCA E9*	*PI3KCA E20*	*ERBB2*	P53	ARID1A
11	32	I a	WT	WT	WT	WT	WT	Normal	Normal
12	39	I c	WT	WT	C1636A (Q546K)	WT	WT	Normal	Normal
13	44	III c	WT	WT	WT	WT	WT	Normal	Normal
14	45	I c	WT	WT	C1636A (Q546K)	WT	WT	Normal	Normal
15	45	III c	WT	WT	G1633C (E545Q)	WT	WT	Normal	Normal
16	38	I a	WT	WT	A1634C (E545A)	WT	WT	Positive	Normal
17	25	I a	WT	WT	WT	WT	WT	Normal	Normal
18	48	I a	WT	WT	C1636A (Q546K)	WT	WT	Normal	Normal
19	69	I a	WT	WT	C1636A (Q546K)	WT	WT	Normal	Normal
20	57	I a	WT	WT	C1636A (Q546K)	WT	WT	Normal	Normal
21	66	I c	WT	WT	WT	WT	WT	Normal	Normal

**Table 3 diagnostics-10-00013-t003:** Mutation and immunohistochemical analysis of serous cystadenoma (SCA). WT; Wild Type.

No.	Age	*KRAS*	*BRAF*	*PI3KCA E9*	*PI3KCA E20*	*ERBB2*	P53	ARID1A
22	25	WT	WT	WT	WT	WT	Normal	Normal
23	73	WT	WT	WT	WT	WT	Normal	Normal
24	81	WT	WT	WT	WT	WT	Normal	Normal
25	47	WT	WT	WT	WT	WT	Normal	Normal
26	52	WT	WT	WT	WT	A2384G (Q795R)	Normal	Normal
27	71	WT	WT	WT	WT	WT	Normal	Normal
28	72	WT	WT	WT	WT	WT	Normal	Normal
29	75	WT	WT	A1634C (E545A)	WT	WT	Normal	Normal
30	55	WT	WT	WT	WT	A2384G (Q795R)	Normal	Normal
31	63	WT	WT	WT	WT	WT	Normal	Normal
32	63	WT	WT	WT	WT	WT	Normal	Normal
33	26	WT	WT	WT	WT	WT	Normal	Normal

**Table 4 diagnostics-10-00013-t004:** Mutation and immunohihistochemical analysis in four cases of LSGC accompanied by SBT. WT; Wild Type.

No.	Age	Stage	LGSC/SBT	*KRAS*	*BRAF*	*PI3KCA E9*	*PI3KCA E20*	*ERBB2*	P53	ARID1A
1	37	III c	LGSC SBT	WT WT	WT WT	WT WT	WT WT	A2384G (Q795R) A2384G (Q795R)	Normal Normal	Normal Normal
6	27	I c	LGSC SBT	WT WT	T1796A (V600E) WT	A1634G (E545G) WT	WT WT	A2384G (Q795R) WT	Normal Normal	Normal Normal
8	48	I c	LGSC SBT	WT WT	WT WT	WT WT	WT WT	WT WT	Normal Normal	Normal Normal
10	40	I c	LGSC SBT	WT WT	T1796A (V600E) WT	A1634C (E545A) A1634C (E545A)	WT WT	WT WT	Normal Normal	Normal Normal

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
