# Peer review of "High Frequency of PIK3CA Mutations in Low-Grade Serous Ovarian Carcinomas of Japanese Patients"

_diagnostics, 2019, doi:10.3390/diagnostics10010013_

Round 1
Reviewer 1 Report
Ishibashi et al. reported the genetic variations in the KRAS, BRAF, PIK3CA, and ERBB2 genes in Japanese low-grade serous ovarian carcinomas (LGSCs)/serous borderline tumors (SBTs)/serous cystadenomas (SCAs). They also observed that a lower frequency of KRAS mutations whereas a higher prevalence of PIK3CA mutations was found in patients with LGSC from Japan compared to Western populations, suggesting that the oncogenic driver of LGSC may be different between Japanese and Western countries.
Overall, this manuscript provides a comprehensive investigation of KRAS, PIK3CA, BRAF and ERBB2 mutation status in Japanese LGSCs. This study will be useful to the wider field and is suitable for publication in Diagnosis.
However, the interpretation of the data is sometimes too simplistic, and could also be clearer and more consistent. Furthermore, the DNA-seq datasets should be deposited in a public repository and the accession numbers should be indicated in the manuscript.
Specific comments are listed below
Introduction: it would be great if the authors update the reference regarding cancer statistics in 2019 (Siegel et al., CA Cancer J Clin, 2019). Results: It is informative to show the different genetic variants of these genes in LGSCs/SBTs/SCAs. Are these gene mutations also related to the overall survival or progression-free survival of LGSC patients? Discussion p.5 line 100: it would be much better to say “How does this discrepancy in PIK3CA mutation occur in patients with LGSC among different studies?” rather than “Why is there this discrepancy in PIK3CA mutation prevalence in LGSC among studies?” Discussion states, "Taken together, the results of the current and previous reports also suggest carcinogenesis of Japanese LGSCs may be different from that of Western LGSCs” and “Based on the current findings, we hypothesize that the main oncogenic signaling pathway in Japanese LGSCs is PIK3CA/AKT, whereas that in Western LGSCs is KRAS/BRAF/ERK”. AKT was not manipulated in this study, so this conclusion cannot be made. Did AKT or NRAS mutation present a higher or lower frequency in the patients with LGSCs from Japan than that in the Western populations? Further, it would be better to make a table to summarize the relevant studies regarding KRAS, PIK3CA, BRAF, and ERBB2 mutation status in LGSCs in both Japan and Western countries. All raw DNA seq data and, preferably, processed data with DNA mutation need to be made publicly available (at the time of manuscript publication) through a database such as NCBI SRA. The location and accession number of DNA seq datasets should be noted in Methods. This manuscript needs to be corrected carefully and rephrased by a native English speaker. There are numerous grammar and wording errors throughout the text that need to be corrected. A few examples are listed as follows:p.1 line 19, p.2, line 50: “P53” should be “p53”
p.1 line 27: “16~54%” should be 16-54%
p.1 line 35: “Ovarian cancer is the leading cause of death due to gynecologic malignancies in the world” should be “Ovarian cancer is the leading cause of death from gynecologic malignancies in the world”
p.1 line 37, p.6 line 115, p.6 line 116: “p53” should be “TP53”
p.1 line 40: “western countries” should be “Western countries”
p.2 line 46 and line 49: “HGSCs” should be “LGSCs”
p.2 line 54: it would be much better to say “All LGSC, SBT, and SCA cases were KRAS wild-type” rather than “KRAS was the wild type in all LGSC, SBT, and SCA cases”
p.2 line 58: “wild type” should be “wild-type”
p.2 line 62: it would be much better to say “We also analyzed mutation status of KRAS, PIK3CA, BRAF, and ERBB2 in LGSCs….” “We also analyzed mutations in the four genes in LGSCs….”
Author Response
Individual Replies to Reviewer 1:
We are grateful for your comments. Our specific answers are listed below.
Comment 1) Ishibashi et al. reported the genetic variations in the KRAS, BRAF, PIK3CA, and ERBB2 genes in Japanese low-grade serous ovarian carcinomas (LGSCs)/serous borderline tumors (SBTs)/serous cystadenomas (SCAs). They also observed that a lower frequency of KRAS mutations whereas a higher prevalence of PIK3CA mutations was found in patients with LGSC from Japan compared to Western populations, suggesting that the oncogenic driver of LGSC may be different between Japanese and Western countries.
Overall, this manuscript provides a comprehensive investigation of KRAS, PIK3CA, BRAF and ERBB2 mutation status in Japanese LGSCs. This study will be useful to the wider field and is suitable for publication in Diagnosis.
However, the interpretation of the data is sometimes too simplistic, and could also be clearer and more consistent. Furthermore, the DNA-seq datasets should be deposited in a public repository and the accession numbers should be indicated in the manuscript.
Response 1) Thank you for your suggestion. Accordingly, we have modified our manuscript throughout. We are going to deposit our sequence data in a public repository. However, for minor revisions, the editorial board only gave us 5 days, and thus we could not finish it. After publication, we will deposit our data into a Japanese DNA sequence data center.
Comment 2) Introduction: it would be great if the authors update the reference regarding cancer statistics in 2019 (Siegel et al., CA Cancer J Clin, 2019).
Response 2) Thank you for your suggestion. I have cited this article (please see reference list).
Comment 3) Results: It is informative to show the different genetic variants of these genes in LGSCs/SBTs/SCAs. Are these gene mutations also related to the overall survival or progression-free survival of LGSC patients?
Response 3) We agree with your opinion. We have analyzed the relationship between genetic variants and patient prognosis, but no association was found. This may be a reflection of the small sample size.
Comment 4)  Discussion p.5 line 100: it would be much better to say “How does this discrepancy in PIK3CA mutation occur in patients with LGSC among different studies?” rather than “Why is there this discrepancy in PIK3CA mutation prevalence in LGSC among studies?”
Response 4) Thank you for your suggestion. We have changed the sentence accordingly (please see page 5, line 100).
Comment 5) Discussion states, "Taken together, the results of the current and previous reports also suggest carcinogenesis of Japanese LGSCs may be different from that of Western LGSCs” and “Based on the current findings, we hypothesize that the main oncogenic signaling pathway in Japanese LGSCs is PIK3CA/AKT, whereas that in Western LGSCs is KRAS/BRAF/ERK”. AKT was not manipulated in this study, so this conclusion cannot be made. Did AKT or NRAS mutation present a higher or lower frequency in the patients with LGSCs from Japan than that in the Western populations? Further, it would be better to make a table to summarize the relevant studies regarding KRAS, PIK3CA, BRAF, and ERBB2 mutation status in LGSCs in both Japan and Western countries.
Response 5) We have not analyzed AKT or NRAS mutation in this study; we plan to do so in a future study. We mean PIK3CA/AKT signaling, not AKT gene alteration. In fact, AKT mutation is very rare in gynecologic tumors.
Following your suggestion, we have created a new table (please see supplementary Table S2 and page 5, lines 99-100).
Comment 6) All raw DNA seq data and, preferably, processed data with DNA mutation need to be made publicly available (at the time of manuscript publication) through a database such as NCBI SRA. The location and accession number of DNA seq datasets should be noted in Methods.
Response 6) We are going to deposit our sequence data to a public repository. However, for minor revisions, the editorial board only gave us 5 days, and thus we could not finish it. After publication, we will deposit our data into a Japanese DNA sequence data center.
Comment 7) This manuscript needs to be corrected carefully and rephrased by a native English speaker. There are numerous grammar and wording errors throughout the text that need to be corrected.
Response 7) Following your suggestion, our manuscript was edited by a native English speaker.
Comment 8) p.1 line 19, p.2, line 50: “P53” should be “p53”
p.1 line 27: “16~54%” should be 16-54%
Response 9) We have changed the text accordingly.
Comment 10) p.1 line 37, p.6 line 115, p.6 line 116: “p53” should be “TP53”
Response 10) We have changed the text accordingly.
Comment 11) p.1 line 40: “western countries” should be “Western countries”
Response 11) We have changed the text accordingly.
Comment 12) p.2 line 46 and line 49: “HGSCs” should be “LGSCs”
Response 12) We have changed the text accordingly.
Comment 13) p.2 line 54: it would be much better to say “All LGSC, SBT, and SCA cases were KRAS wild-type” rather than “KRAS was the wild type in all LGSC, SBT, and SCA cases”
Response 13) We have changed the sentence.
Comment 14) p.2 line 58: “wild type” should be “wild-type”
Response 14) We have changed it.
Comment 15) p.2 line 62: it would be much better to say “We also analyzed mutation status of KRAS, PIK3CA, BRAF, and ERBB2 in LGSCs….” “We also analyzed mutations in the four genes in LGSCs….”
Response 15) We have changed the sentence.
Reviewer 2 Report
The goal of this study was stated as "In the current study, we evaluated the prevalence KRAS, BRAF, PIK3CA, and 49 ERBB2 mutations in Japanese HGSCs and clarified the genetic drivers of these mutations". however the results and conclusions are indicating differently: "the mutation frequency of KRAS in LGSCs/SBTs 155 in Japan is lower than that in Western countries".
Please check for consistency in hypothesis and conclusions
Author Response
Reply to Reviewer 2:
We are grateful for your comments. Our specific responses are listed below.
Comment 1) The goal of this study was stated as "In the current study, we evaluated the prevalence KRAS, BRAF, PIK3CA, and 49 ERBB2 mutations in Japanese HGSCs and clarified the genetic drivers of these mutations". however the results and conclusions are indicating differently: "the mutation frequency of KRAS in LGSCs/SBTs 155 in Japan is lower than that in Western countries".
Please check for consistency in hypothesis and conclusions
Response 1) We are grateful for your comments. Following your suggestion, we added the following comment to the revised manuscript: “In the current study, we evaluated the prevalence of KRAS, BRAF, PIK3CA, and ERBB2 mutations in Japanese LGSCs, not only clarifying the genetic drivers of these mutations but also the difference in mechanisms of carcinogenesis between Japanese and European LGSCs.” (please see page 2, lines 48–50).